# Sexualized Drug Use and Chemsex among Men Who Have Sex with Men in Europe: A Systematic Review and Meta-Analysis

**DOI:** 10.3390/jcm13061812

**Published:** 2024-03-21

**Authors:** Marina Coronado-Muñoz, Emilio García-Cabrera, Angélica Quintero-Flórez, Encarnación Román, Ángel Vilches-Arenas

**Affiliations:** 1Preventive Medicine and Public Health Department, Faculty of Medicine, University of Seville, Av. Sanchez Pizjuan s/n, 41009 Seville, Spain; marcormun2@alum.us.es (M.C.-M.); aquintero@us.es (A.Q.-F.); encarnacion.roman.sspa@juntadeandalucia.es (E.R.); ava@us.es (Á.V.-A.); 2Department of Preventive Medicine, Hospital Universitario de Valme, Ctra. de Cádiz Km. 548.9, 41014 Seville, Spain; 3Department of Preventive Medicine, Virgen Macarena University Hospital, Avda. Dr. Fedriani 3, 41009 Seville, Spain

**Keywords:** chemsex, unsafe sex, substance-related disorders, drug dependence, sexually transmitted diseases, sexual and gender minorities, men who have sex with men

## Abstract

**Background:** In recent years, in Europe, there has been a growing concern about the use of sexualized drugs among men who have sex with men (MSM), due to its possible link to an increase in sexually transmitted infections. The aim of this review is to study the prevalence of chemsex, and the sexualized drug used in Europe, describing both different consumption patterns and other sexual behaviors considered risky and their possible relationship with positivity in diagnoses of sexually transmitted infections, including human immunodeficiency virus. **Methods**: We conducted a literature review in the main scientific databases (PubMed, Embase, Scopus, Cochrane Library, Web of Science), filtering for articles published between January 2018 and April 2023 that collect information on sexualized drug use and sexual practices conducted in European countries among men who have sex with men, including whether these behaviors can lead to diagnoses of sexually transmitted infections. **Results:** The definition of drugs included in chemsex is not clearly defined and shows heterogeneity between study publications; the three drugs presented in all manuscripts are mephedrone, GHB/GBL, and crystal methamphetamine. The prevalence of chemsex in Europe is 16% [11–21%] among MSM. The most frequent risky sexual behavior associated with chemsex practice was unprotected sex with a high number of partners. The log risk ratio of STIs was 0.86 (95% CI: 0.49 to 1.23). **Conclusions:** Adherence to definitions, stringent research methodologies, and focused interventions are needed to tackle the intricate relationship between substance use, sexual behavior, and the risk of HIV/STI transmission in MSM.

## 1. Introduction

The use of sexualized drugs (SDU) is defined as the use of a wide range of illicit substances in the context of sexual relationships. Within SDU, there is a subcategory known as “chemsex”. The differentiation of this subgroup lies in the drugs used, which are consumed by men immediately before or during sexual relations with other men [1], often leading to prolonged sexual sessions and distinct characteristics in terms of other sexual practices. The consumption of these drugs may be linked to the search for multiple effects, such as facilitating, improving, or prolonging sexual relationships [1,2]. In recent years, the practice of “chemsex” has emerged as a public health problem, possibly resulting from the increasing number of users who access and utilize geosocial networks and applications for this type of sexual practice. The concept is socially grounded, dependent on user preferences, trends, and drug availability, causing it to vary over time and between different countries or regions [2,3].

This practice is widely studied in the UK [1,2,3] where the public health authority in 2015 reported a list of the drugs included in chemsex practice [4]. However, a certain definition of drugs that are included in the practice of chemsex is not established. SDU appears to be more associated with sporadic use, occurring weekly or monthly [5]. Most of the literature divides different substances in SDU into different categories based on the effect search [1,5,6,7,8,9,10]. They include substances to improve sexual performance, such nitrile derivates, also called “poppers”, and Viagra [9,10,11], recreational or party drugs (such as ecstasy, amphetamines, cocaine, and ketamine), and drugs associated with “chemsex” (such as methamphetamine, mephedrone, and GHB/GBL) [3,12,13]. It is important to mention “slamsex” (also known as “rough sex”), a term that has emerged in recent years and is closely associated with the gay population. “Slamsex” is a specific form of drug use and sex involving intravenous drug injection for sexual activities, using the same drugs as “chemsex” (typically methamphetamine, GHB/GBL, and mephedrone). This practice raises concerns about potential health consequences, particularly in terms of HIV and hepatitis C transmission [14,15]. 

It is assumable to establish a risk ladder among different SDU patterns, involving both HIV/sexualized transmitted infection (STI) diagnoses and risky sexual behaviors influenced by drug effects, including group sex, transactional sex, unprotected anal sex, and multiple sexual partners [1,3,8,16,17]. As a result, SDU partially causes bacterial STIs and HCV or serves as an indirect indicator of other risk components.

On another note, it is crucial to remember that HIV remains a global health issue, especially among the MSM population. Despite the progress made with pre-exposure prophylaxis (PrEP), a false sense of confidence has led to the neglect of barrier methods during sexual relations, resulting in an increase in other STIs such as gonorrhea, syphilis, hepatitis, urethritis, chlamydia, and genital warts [11,18]. The prevalence of “chemsex” among HIV-infected MSM is high, and conversely, the prevalence is lower among stable partners [8,16,18].

Our study aims to understand the real prevalence of the problem, and characterizing these behaviors can contribute to improving sexual health and strengthen the prevention of sexually transmitted infections, including HIV. Considering all of the above, in our systematic review, our objective is to answer the following research questions. Is there a single definition of chemsex in terms of the pattern of consumption and the type of drugs used? What is the actual magnitude of the problem in Europe? Is this practice truly related to other risky behaviors and does it pose an increased risk of HIV and other sexually transmitted diseases? Our objectives are to examine the patterns of sexualized drug use in the population of men who have sex with men in Europe and to analyze whether sexualized drug use, along with sexual practices that may be considered risky a priori, is related to a potential increase in the diagnoses of STIs in the mentioned population.

## 2. Materials and Methods

This systematic review adheres to the Preferred Reporting Items for Systematic Reviews and Meta-Analyses (PRISMA) guidelines [19]. The protocol was registered in the database of the International Prospective Register of Systematic Reviews (PROSPERO) after meeting its inclusion criteria [20], registration number CRD42024482623.

### 2.1. Data Sources and Searchers 

The review of the literature spanned from January 2022 to April 2023. We conducted searches of five electronic databases, namely MEDLINE through PubMed, Cochrane, Embase, Web of Science, and Scopus, to identify pertinent articles. The search strategy was formulated during a panel meeting following an initial article search. It involved employing key phrases and/or their abbreviations according to a metadata system (MeSH) and various combinations of these phrases to enhance search efficiency. The search strategy is included in Appendix A. A comprehensive set of terms to characterize the intended population was developed using the PICO acronym as a guide:P (population): men who have sex with men;I (intervention): “chemsex”, “sexualized drug use”;C (comparator): men who have sex with men;O (outcomes): “sexually transmitted diseases”.

The titles and abstracts of scientific articles retrieved from the databases were analyzed for inclusion criteria. The inclusion criteria were as follows: (1) articles published between January 2018 and April 2023; (2) articles written in English, Spanish, and French; (3) articles corresponding to experimental, or observational studies; (4) articles that address the following research PICO question: “Does SDU truly influence the increase in the diagnosis of sexually transmitted infections?”. Articles that focus their main objective on psychological aspects such as depressive disorders and topics related to the field of mental health were excluded. Studies conducted outside of European countries were also excluded. Reference lists within the reviewed publications were also examined to ensure that no additional articles met the inclusion and exclusion criteria.

### 2.2. Study Selection

The examination of the gathered papers underwent a five-stage process. Initially, articles were searched, followed by the removal of duplicates as the second step. The third step involved reviewing the titles and the fourth step involved reviewing the abstracts of papers deemed potentially relevant to the research questions. Subsequently, the full texts of the articles identified in the initial selection were thoroughly reviewed and their quality assessment was performed. Throughout all stages, two independent reviewer teams (M.C.-M. and E.G.-C; E.R. and A.Q.-F) performed the review, with a third independent reviewer (Á.V.-A.) stepping in for contentious cases.

### 2.3. Data Extraction 

Two reviewers (E.G.-C. and A.Q.-F.) independently extracted and documented data from each included study according to the recommendations of the Centre for Reviews and Dissemination [21]. We extracted data including the year of publication, country, study design, quality assessment, sample size, number of chemsex users, number of SDU users, outcome measures, and study results.

### 2.4. Methodological Quality Assessment

The methodological quality of observational studies was assessed using the STROBE guideline [22]. All studies with 50% or less checked items were excluded for review. In all cases, the evaluation was conducted by two independent reviewers.

### 2.5. Data Synthesis and Analysis

We extracted for all articles the absolute frequency of SDU user and chemsex user, following the definition included in the articles selected, and the sample size for the calculation of the prevalence of SDU and chemsex. We also extracted HIV diagnostics and the risky behaviors associated with chemsex practice. A meta-analysis was performed for the estimation of the prevalence of chemsex and the risk ratio of STIs. Two random-effects models were fitted to the data. The amount of heterogeneity was estimated using the DerSimonian–Laird estimator [23]. In addition to the estimate of tau^2^, the Q-test for heterogeneity [24] and the I^2^ statistic are reported. In case any amount of heterogeneity is detected (tau^2^ > 0, regardless of the results of the Q-test), a prediction interval for the true outcomes is also provided. Studentized residuals and Cook’s distances are used to examine whether studies may be outliers and/or influential in the context of the model. Studies with a studentized residual larger than the 100 × (1 − 0.05/(2 Xk))th percentile of a standard normal distribution are considered potential outliers (i.e., using a Bonferroni correction with a two-sided alpha of 0.05 for k studies included in the meta-analysis). Studies with a Cook’s distance larger than the median plus six times the interquartile range of the Cook’s distances are influential. All statistical analysis were performed using Jamovi version 2.3.21 [25,26], and the meta-analysis was performed using the metafor package [27].

## 3. Results

We obtained 905 scientific articles that include information on SDU, sexual practices, and sexually transmitted infections in the MSM population. A total of 470 (51.9%) were eliminated after the first filter, and of the 435 remaining, 17 articles were read to evaluate the selection criteria established. Finally, eight articles passed the final quality filter and constitute the main result (Figure 1). All the included articles were cross-sectional studies performed across Europe, in 16 different countries; three of them were multinational studies [28,29,30,31,32,33,34,35]. The median sample size of the different studies was 2883 MSM, ranging from 250 to 9407, and four of eight articles (50%) were obtained by attendants in sexual clinics [29,30,31,34]. The main results and conclusion of the eight articles are included in the Appendix A. 

### 3.1. Chemsex Definition 

The drugs included in the chemsex definition varied in the included articles. The drugs included in all manuscripts were gamma hydroxybutyrate or gamma butyrolactone (GHB/GBL)—also named liquid ecstasy—mephedrone, and crystal meth. Ketamine is another drug generally included in chemsex, included in five of the eight manuscripts [28,29,30,31,34]. The complete list of the drugs included in the chemsex definition is included in Table 1.

### 3.2. Prevalence of Chemsex and SDU

The analysis encompassed eight studies. The prevalence of chemsex observed in these studies varied from 0.03 to 0.26. Using a random-effects model, the estimated prevalence was 16% (95% CI: 11.1% to 20.9%), indicating a significant deviation from zero (z = 6.37, *p* < 0.001). The Q-test suggested heterogeneity in the true outcomes (Q7 = 1697, *p* < 0.0001, tau^2^ = 0.07, I^2^ = 99.59%). Figure 1 illustrates the prevalence of chemsex across the eight studies along with their respective weights in the model.

### 3.3. Sexual Risky Behaviors 

The leading risky sexual behavior reported was open live sex, which means a high number of partners in the study period, together with unprotected anal intercourse. Table 2 describes all the risky sexual behaviors associated with chemsex, as well as the heterogenicity of the results, presented as proportions, median, odds ratio, and prevalence ratio, which did not allow for a deeper analysis. 

### 3.4. Sexually Transmitted Disease Risk

Eight studies were included in the analysis. The observed log risk ratios ranged from 0.236 to 1.496. The estimated log risk ratio based on the random-effects model was 0.86 (95% CI: 0.49 to 1.23). Therefore, the average outcome differed significantly from zero (z = 4.58, *p* < 0.0001). According to the Q-test, the true outcomes appeared to be heterogeneous (Q_7_ = 96.98, *p* < 0.0001, tau^2^ = 0.2526, I^2^ = 92.78%). An examination of the studentized residuals revealed that none of the studies had a value larger than ± 2.7344 and hence there was no indication of outliers in the context of this model. According to the Cook’s distances, none of the studies could be overly influential. Neither the rank correlation nor the regression test indicated any funnel plot asymmetry (*p* = 0.3988 and *p* = 0.3417, respectively). Figure 2 presents the forest plot of the model. 

## 4. Discussion

One of the objectives of our work has been to establish the prevalence of chemsex and the difference in the sexualized use of drugs in Europe. The main difficulty we have encountered is that there is no universal and accepted definition of chemsex. Although all articles include mephedrone, crystal methamphetamine, and GHB/GBL [28,29,30,31,32,33,34,35], others also include ketamine [28], drugs such as so-called “club drugs” or drugs to enhance sexual effects such as poppers and Viagra, and other illicit drugs such as cocaine, ecstasy, or cannabis [29,30,31,34]. This has already been reported in previous reviews [5,36], which is why a consensus has been developed in the public health system of England [4], which is a reference in Europe because it is the region where the phenomenon of chemsex has been studied the most. Despite having a stable definition, this has only been followed in the subsequent literature by a minority proportion of the studies in our article 3/8 (37.5%). The importance of adhering to the established definition and only reporting usage will allow us to understand the magnitude of the problem in more detail and establish specific measures to address this and the more serious issue of injected drug use, such as slamsex [14,37], where there has been a greater increase according to the latest drug report in Europe [38]. We have established that the prevalence of chemsex is between 9% and 21% in Europe. These data are consistent with other reviews [36] and are closer to other previous reviews [5].

In our study, the risky sexual practices primarily associated with chemsex that we have identified are engaging in unprotected sex and having a high number of encounters. These findings have been extensively supported by other reviews [5,6,36]. We were unable to extract specific data on this increased risk due to the high heterogeneity documented in these results. The fact that these two practices are found to be mainly associated with chemsex and not others is consistent with the use of these drugs and their ubiquity, as well as the established social norms within MSM circles [39,40].

Despite the reported increase in risk associated with risky practices related to chemsex documented in all studies and noted in previous reviews [5,6,36], we did not find a clear increase in risk in our results. This may be due to two fundamental issues. The first is the heterogeneity of the samples from which we derived the results of our study. One particular question of this is the heterogenicity observed in the studies and the different period of evaluation ranging from six months to one year. This changes the total number of risky sexual behaviors and STIs. The other is the correlation with the higher usage of pre-exposure prophylaxis (PrEP) and post-exposure prophylaxis (PEP) as demonstrated by other authors [39,40,41,42]. Although this would only be for HIV; other sexually transmitted infections (STIs) are not covered by prophylaxis, but these are less prevalent and are considered differently in various manuscripts.

In our systematic review and meta-analysis, we have identified the following weaknesses. The first refers to the predominantly cross-sectional nature of research on the sexualized use of drugs. Of the 421 articles remaining after removing duplicates in the selection process, only 17 were analytical studies, none of which passed the successive selection criteria. Hence, caution must be exercised in interpreting the results of the risk factor.

The second weakness lies in the significant heterogeneity observed, as previously mentioned, in the meta-analysis results, both in the prevalence of chemsex and in the risk of sexually transmitted diseases. This heterogeneity is due to variations in sample selection. Some of the studies included in this review [29,30,31,34] extracted data from sexually transmitted infection clinics, which can increase the prevalence of HIV/STIs. Others extracted HIV/STI status data through self-reported questionnaires [28,32,33,35], which could lead to an underestimation of prevalence. 

Furthermore, in self-reported surveys, when asking about behaviors, consumption, or practices considered risky, respondents may feel more disinhibited and answer candidly, and in contrast could be affected by complacency bias. If these surveys emphasize studying variables in the last 6 months or within the past year (among others), then recall bias may occur.

Regarding the point concerning male sex workers, it is noteworthy that several articles including male sex workers (or at least specifying their inclusion) were found but were excluded during the selection process.

It is important to note that this study includes articles examining MSM populations engaged in SDU practices in specific locations, such as Spain or the Netherlands. However, as with the inclusion of populations from more specific territories in our review, as strengths of the study, we mention the analysis of studies that include populations from various European countries, major cities, and residents of neighboring regions or areas adjacent to cities, thus forming a heterogeneous population that, in our view, may be representative of the overall population. Furthermore, discussing other strengths, most of the studies were conducted using a large sample of participants. Another stretch is the quality assessment of the included manuscript based on the standardized score.

## 5. Conclusions

Our study contributes to understanding chemsex and the associated behaviors among MSM in Europe, highlighting the need for standardized definitions, rigorous research methodologies, and targeted interventions to address the complex interplay between substance use, sexual behavior, and HIV/STI transmission risk in this population.

## Data Availability

No new data were created or analyzed in this study. Data sharing is not applicable to this article.

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
