# Peer review of "Sexualized Drug Use and Chemsex among Men Who Have Sex with Men in Europe: A Systematic Review and Meta-Analysis"

_jcm, 2024, doi:10.3390/jcm13061812_

Round 1

Reviewer 1 Report

Comments and Suggestions for Authors

The manuscript is a metanalysis on the prevalence of SDU, risky behaviours among MSM in Europe. A total of 8 studies have been selected through a selection procedure. Inclusion criteria are clear as well as is the selection process (5 steps by 2 reviewers). The quality assessment was performed according to the STROBE guidelines. The study suggests that the prevalence of SDU is 16% (11-21%), the most frequent risky behaviours associated to chemsex are unprotected sex and high number of partners, and the log risk ratio of STIs being 0.86 (95%CI: 0.49- 1.23). The English language is clear and the data are well presented. However, some minor issues need to be addressed by the authors. Heterogeneity of studies is the major limitation to draw correct conclusions about the link with STIs from this analysis. The authors need to focus on the main end point (and clearly specify it in the method section): i.e. prevalence of SDU and risky behaviours, which means that studies need to be homogeneous in terms of type of investigation (a questionnaire? Clinicians’ perceptions? Systematic questioning or what? for both chemsex use and risky behaviours). Some info about the tools for data collection in all 8 studies may be useful. In addition, the frequency of STIs depends largely on similarity in screening frequency and completeness of investigations (whether it is systematic or based on symptoms only). Duration of follow up may also change the absolute number of STIs. Without homogeneity in the collection of information, no clear and precise link can be identified between risky behaviours and STIs. I would suggest to elaborate such reasoning in the period between line 231-238 which explains why the study cannot assess a link between chemsex-risky beh-STIs appropriately In such a perspective, not including STIs in the title does make sense to me, but wording in the text should follow the same caution. In the conclusions section, STIs are not mention correctly Minor issues • For clarity I would include for each study in table 1 and figure1 the number of patients so as to rapidly identify the magnitude • In table 2, in the legend, the acronymous CAR may be incorrect (Insertive, receptive?)

Author Response

  1. The manuscript is a metanalysis on the prevalence of SDU, risky behaviours among MSM in Europe. A total of 8 studies have been selected through a selection procedure. Inclusion criteria are clear as well as is the selection process (5 steps by 2 reviewers). The quality assessment was performed according to the STROBE guidelines. The study suggests that the prevalence of SDU is 16% (11-21%), the most frequent risky behaviours associated to chemsex are unprotected sex and high number of partners, and the log risk ratio of STIs being 0.86 (95%CI: 0.49- 1.23). The English language is clear and the data are well presented. However, some minor issues need to be addressed by the authors. Heterogeneity of studies is the major limitation to draw correct conclusions about the link with STIs from this analysis.
  2. Thank you very much for your constructive resume of the work. On behalf of all authors, I am very grateful for your time and dedication in reviewing our manuscript. Your constructive feedback gives us the right impulse to improve the quality and impact of our In general, in Table T1 of the supplemental material there are some of the information that reviewer request, and we strongly agree with the review that some of this information is essential to understand the main limitation of our manuscript, the heterogenicity of publications. Now, we address all your comments one by one.
  3. The authors need to focus on the main end point (and clearly specify it in the method section): i.e. prevalence of SDU and risky behaviours, which means that studies need to be homogeneous in terms of type of investigation (a questionnaire? Clinicians’ perceptions? Systematic questioning or what? for both chemsex use and risky behaviours). Some info about the tools for data collection in all 8 studies may be useful. In addition, the frequency of STIs depends largely on similarity in screening frequency and completeness of investigations (whether it is systematic or based on symptoms only).
  4. Thank for the comment. The main endpoint of our manuscript is the evaluation of the STIs such as that the review could check in the PICO question and in the PROSPERO registry. We strongly agree with the data and the information on tool evaluation is basic for the study, and you can find that information in the supplemental material table.

Duration of follow up may also change the absolute number of STIs. Without homogeneity in the collection of information, no clear and precise link can be identified between risky behaviours and STIs. I would suggest to elaborate such reasoning in the period between line 231-238 which explains why the study cannot assess a link between chemsex-risky beh-STIs appropriately In such a perspective, not including STIs in the title does make sense to me, but wording in the text should follow the same caution.

  1. Thank you for the comment, we agree with the appreciation of the reviewer and include the different number of STI and risky behaviors due to the different follow-up period included in the studies. In lines 233-236

The first is the heterogeneity of the samples from which we derived the results of our study. One particular question of this is the heterogenicity observed in the studies and the different period of evaluation ranging from 6 moths to one year. This changes the total number of risky sexual behaviors and STI.

In the conclusions section, STIs are not mention

  1. Thank you for your comment. We include in the discussion that further studies are needed to address correctly the interplay between chemsex, sexual risky behaviors, and HIV and another sexually transmitted infection.

Minor issues • For clarity I would include for each study in table 1 and figure1 the number of patients so as to rapidly identify the magnitude • In table 2, in the legend, the acronymous CAR may be incorrect (Insertive, receptive?)

  1. Thank you again for your appreciation. We include in Table 1 the sample size of each studies, in Figure 1 and 2 we cannot include it because it is a plot generated for the statistical software Jamovi and couldn’t be modified. We also correct the legend of Table

Thank you again for your dedication to this review process, I the reviewer needs any further information, please do not hesitate to contact with me.

Best Regards.

Reviewer 2 Report

Comments and Suggestions for Authors

Below I present my observations and comments on the manuscript (you will find the same ones as notes in the writing):

Line 93.- Which were the criteria to specific use of these databases?

Line 105.- Could you describe the reasons or provided further details about the evaluated years range?

Line 116.- Did you use any software to remove the duplicates or was a manual work?

Line 126.- Regarding the collected variables, the mentioned ones were the uniques (a priori defined) or the conserved (most interesting) after filter many others?

Line 130.- So, only included observational studies? Or what did you mean with experimental studies? Please review line 107.

Line 130/2.- The STROBE guideline is specific to observational studies or was an adaptation to accomplish your objective (quality assessment of observational “cross-sectional” studies)?

Line 165.- Did you evaluated the combination of drugs?

Line 186.- In a more specific way, did you evaluate sexual rol (receptive or insertive)?

Line 219.- I consider important to offer some details about the injected drugs (to highlight the relevance)… is there have relation with the transmission way (sexual and blood)?

Line 236.- For all your included studies was clear the HIV status of all participants? I mean, were you capable of address two different populations within the MSM in your studies collection (HIV positives and HIV negatives)?

Line 240.- Systematic review...

Line 244.- I agree with this reasoning, having a transversal nature we cannot assume the calculation of “risk” as is.

Line 253.- In contrast, the data might be was/were affected by complacency bias…

Author Response

  1. Below I present my observations and comments on the manuscript (you will find the same ones as notes in the writing):
  2. On behalf of all authors, I give our sincerest grateful for your time and effort to improve our manuscript. Now, I address all your comments one by one.
  3. Line 93.- Which were the criteria to specific use of these databases?

  1. The database selection was used because PubMed, EMBASE and Cochrane are the three top databases for clinical research, and Scopus and Web of Science are the two best interdisciplinary
  2. Line 105.- Could you describe the reasons or provided further details about the evaluated years range?
  3. The previous published systematic review was published in 2018 and we can evaluate the new studies that were not included in the previous review.
  4. Line 116.- Did you use any software to remove the duplicates or was a manual work?

  1. All result of research were exported to csv files and merged into an Excel database. In Excel we filter the duplicate by title.

  1. Line 126.- Regarding the collected variables, the mentioned ones were the uniques (a priori defined) or the conserved (most interesting) after filter many others?

  1. The mentioned variables are the unique ones, because are the needed for our objectives. You could evaluate in Table T1 in the Supplementary Material.

  1. Line 130.- So, only included observational studies? Or what did you mean with experimental studies? Please review line 107.

  1. In our inclusion criteria are included observational and experimental studies, but we only find observational studies that address the five-stage of selection process.

  1. Line 130/2.- The STROBE guideline is specific to observational studies or was an adaptation to accomplish your objective (quality assessment of observational “cross-sectional” studies)?

  1. STROBE guidelines have any specific criteria for each observational studies, the reviewer could check at website https://www.strobe-statement.org/checklists/.

  1. Line 165.- Did you evaluated the combination of drugs?

  1. Thank you for your interesting comment, but due to the kind of studies that resulted, it was impossible to evaluate the different combinations of drugs because we could select sub-samples in the study.

  1. Line 186.- In a more specific way, did you evaluate sexual rol (receptive or insertive)?

  1. In Table two, the risky sexual behaviors are considerate like condomless anal receptive (CAR) and condomless anal insertive (CAI).

  1. Line 219.- I consider important to offer some details about the injected drugs (to highlight the relevance) … is there have relation with the transmission way (sexual and blood)?

  1. Line 236.- For all your included studies was clear the HIV status of all participants? I mean, were you capable of address two different populations within the MSM in your studies collection (HIV positives and HIV negatives)?

  1. Yes, in all included studies are defined patients exposed to chemsex and with HIV/STI and patient without chemsex exposition and HIV/STI diagnosed in previous 6 months or year.

  1. Line 240.- Systematic review...

  1. Thank you, we correct it.

  1. Line 244.- I agree with this reasoning, having a transversal nature we cannot assume the calculation of “risk” as is.

  1. Is the main limitation of our results, cohorts of chemsex users are needed to stablish the real risk.

  1. Line 253.- In contrast, the data might be was/were affected by complacency bias…

  1. Thank you for your appreciation, we include it in the text.

Thank you again for your dedication to this review process, I the reviewer needs any further information, please do not hesitate to contact with me.

Best Regards.